# Methodological Aspects of Green Extraction of Usnic Acid Using Natural Deep Eutectic Solvents

**DOI:** 10.3390/molecules28145321

**Published:** 2023-07-10

**Authors:** Magdalena Kulinowska, Sławomir Dresler, Agnieszka Skalska-Kamińska, Agnieszka Hanaka, Maciej Strzemski

**Affiliations:** 1Department of Analytical Chemistry, Medical University of Lublin, 20-093 Lublin, Poland; kulinowskamagdalena@gmail.com (M.K.); slawomir.dresler@umlub.pl (S.D.); agnieszka.skalska-kaminska@umlub.pl (A.S.-K.); 2Department of Plant Physiology and Biophysics, Institute of Biological Sciences, Maria Curie-Skłodowska University, 20-033 Lublin, Poland; agnieszka.hanaka@poczta.umcs.lublin.pl

**Keywords:** *Cladonia uncialis*, natural products, lichens, green chemistry, design of experiments

## Abstract

Usnic acid (UA) is a compound with multiple biological activities that make it useful in various industries, e.g., pharmaceutical, cosmetic, dentistry, and agricultural sectors. Lichens are the primary source of UA, which is primarily extracted using acetone. This study aimed to investigate the solubility of UA in numerous natural deep eutectic solvents (NADESs) and use a mixture of thymol and camphor as a NADES in the optimization of the UA extraction process with the design of experiments method. For numerical optimization, the following parameters were employed in the experiment to confirm the model: a camphor-to-thymol ratio of 0.3, a liquid-to-solid ratio of 60, and a time of 30 min. The obtained experimental results aligned well with the predicted values, with the mean experimental value falling within the confidence interval, exhibiting deviations between 11.93 and 14.96. By employing this model, we were able to optimize the extraction procedure, facilitating the isolation of approximately 91% of the total UA content through a single extraction, whereas a single acetone extraction yielded only 78.4% of UA.

## 1. Introduction

Usnic acid (UA) is a phenolic compound with a dibenzofuran structure occurring in the form of two enantiomers (Figure 1) [1] in several lichen genera (*Alectoria*, *Cladonia*, *Evernia*, *Lecanora*, *Parmelia*, *Ramalina*, *Usnea*, and *Xanthoparmelia*) as a yellow cortical pigment [2,3,4,5,6]. Nowadays, it is the most extensively investigated lichen secondary metabolite. Its numerous biological activities have been proved, e.g., antiprotozoal, anti-inflammatory, antimicrobial, antiviral, analgesic, insecticidal [7], antitumor, antipyretic, antiproliferative, antioxidant [8], photoprotective [6], cytoprotective, gastroprotective, and immunostimulatory properties [9,10]. Considering the above-mentioned activities, UA has found application in cosmetology, agriculture, the pharmaceutical industry, and dentistry [3]. It has been used in dietary supplements for weight loss because of the slimming effect observed in humans [11].

However, for UA to be used in industry, it must be isolated from lichen thallus, because the use of a synthetic compound increasingly fails to meet the expectations of modern consumers, does not comply with the tenets of “green chemistry”, and the use of plant raw material requires its standardization and is associated with the introduction of other metabolites into the product as well. The values of the LogP coefficient (2.88) [12] and topological polar surface area (118 Å^2^) [13] evidence the lipophilic nature of this compound, which largely determines its poor solubility in polar solvents. For this reason, UA is extracted from lichens using solvents such as acetone [5,10,14,15,16,17,18,19,20], dichloromethane [21,22,23,24], petroleum ether [3,4,25], n-hexane [26], and heptane [27]. Methanol and ethanol are also used as solvents, although much less frequently [4,28,29], as their extraction yields are lower than those of non-polar solvents. Modern trends aimed at reducing the use of toxic solvents have initiated extensive research into the possibility of “green” extraction of UA. There are reports on extraction with supercritical CO_2_ [2,30] and solvents such as sunflower oil [11], glycerol, rapeseed oil, and olive oil [31].

An alternative to the solvents classically used in phytochemistry is offered by natural deep eutectic solvents (NADESs), which consist of common plant metabolites such as carbohydrates, amino acids, simple organic acids [32], or components of essential oils [33,34]. When mixed at room temperature, these compounds form liquids and provide a medium in which other metabolites can dissolve. Hence, they are considered the third type of solvent in plant cells [35]. Since these solvents are natural and non-toxic, they can be alternative extractants of compounds that have been extracted with highly toxic solvents to date [32,36]. To date, a wide variety of metabolites have been extracted with NADESs, mostly polar [32,37,38,39] but also non-polar compounds [33,40]. Therefore, it seems tempting to explore the possibility of using NADESs in the extraction of UA, especially since there are no references in the literature on this subject to the best of our knowledge. In the present study, an attempt was made to determine the solubility of UA in dozens of NADES compositions and the optimal extraction parameters for efficient extraction of this compound from *Cladonia uncialis* (L.) Weber ex F.H. Wigg. (Figure 2).

## 2. Results and Discussion

### 2.1. Solubility of Usnic Acid

#### 2.1.1. Solubility of Usnic Acid in One-Component Solvents

Table 1 shows the solubility of UA at 20 °C in selected synthetic and natural solvents as well as in lipophilic and hydrophilic NADESs. Among one-component solvents, ethyl acetate and acetone provided good solubility of UA (656.72 and 636.72 mg/100 mL, respectively), which is in agreement with previous papers indicating that these extractants are the most suitable for UA extraction from lichens [14,19,20,23,41]. These values are close to those obtained by O’Neil [42] at a temperature of 25 °C (770 and 880 mg/100 mL for acetone and ethyl acetate, respectively). It was found that the solubility of UA was approx. two times lower in DMSO and as much as fifty times lower in methanol than in acetone and ethyl acetate. Noteworthily, natural solvents, such as levulinic acid and propanediol, are frequently used as ingredients in NADESs. However, the solubility of UA in these compounds (55.50 and 41.74 mg/mL for levulinic acid and propanediol, respectively) did not match the values obtained for acetone or ethyl acetate being from ca. 12 to 16 times lower. By contrast, the solubility of levulinic acid and propanediol were four and three times higher, respectively, than the solubility of UA in methanol.

#### 2.1.2. Solubility of Usnic Acid in Lipophilic NADESs

Among the tested NADES systems, the lipophilic mixtures of thymol with borneol and thymol with camphor (Table 1) turned out to be the best UA solvents, which seems to be justified by the lipophilic nature of UA. The highest solubility, exceeding 1 g/100 mL, was found for a mixture of thymol and borneol mixed in a molar ratio of 3:1; however, this mixture turned out to be unstable and crystallized very quickly. A much more stable NADES was created by mixing camphor and thymol (molar ratio: 1:1, 1:4, 3:2, 3:7). Approximately twice the solubility of UA relative to the camphor:thymol mixtures was recorded for the thymol:lactic acid:water mixtures (319–587 mg/100 mL), while the addition of choline chloride to these components resulted in a dramatic decrease in solubility (8.51 mg/100 mL in the thymol:lactic acid:water:choline chloride mixture in a molar ratio of 1:3.6:2:1, respectively). The solubility of UA in the thymol:menthol mixtures ranged from approximately 300 to 500 mg/100 mL, with a higher thymol content having a positive effect on UA solubility. Interestingly, the solubility of UA in the menthol:lactic acid mixture (molar ratio 1:1.8) was about twice as low as in the thymol:lactic acid:water mixture (molar ratio 1:2.7:1.5). This also indicates the important role of thymol as a component of NADESs used to dissolve UA. NADESs composed of menthol and fatty acids (lauric and myristic acids) yielded UA solutions with concentrations of about 110 mg/100 mL, and it appeared that they had the lowest potential as UA solvents among the lipophilic mixtures tested.

#### 2.1.3. Solubility of Usnic Acid in Hydrophilic NADESs

Hydrophilic NADESs were undoubtedly much worse solvents and showed a significantly lower UA extraction ability compared to lipophilic NADESs (Appendix A). Only a mixture of proline and urea (molar ratio 1:1) yielded a UA solution with a comparable concentration to that in the methanolic solution. The mixtures of proline with levulinic acid and urea with choline chloride provided a UA solubility of ~9 mg/100 mL. All the other hydrophilic NADESs tested only yielded concentrations comparable to those in the solutions with pure lactic acid, glycerol, or water.

### 2.2. Usnic Acid Accumulation in Lichen

*Cladonia uncialis* was used for the determination of UA, and acetone was used as a standard extractant to perform a total UA extraction. However, the exhaustive extraction of UA with acetone required multiple cycles using fresh portions of the extractant, and acetone did not seem to be the best extractant of this compound (compared to the NADESs used in this experiment). Ten-fold extraction with acetone made it possible to determine the content of UA in the *C. uncialis* at the level of 14.5 ± 0.1 mg/g (exhaustive extraction), while three-fold extraction, commonly used in the isolation of plant metabolites, resulted in the isolation of 93.6% of UA (78.4% and 90.4% in the first and second steps, respectively) (exemplary chromatograms are presented in Figure 3). Successive extraction steps increased the yield to 95.2%, 96.8%, and 97.6% for the four-, five-, and six-fold extractions, respectively. The next steps (7, 8, 9, and 10) made it possible to isolate an additional 0.3 ± 0.1 mg/g of UA (2.4% of the total isolated content).

Although the solubility of UA in acetone (see Table 1) allowed exhaustive extraction from the raw material (1.45 mg of UA from 100 mg of raw material with a UA solubility of about 9.55 mg/1.5 mL of acetone), the extraction yield was relatively low, and the subsequent extraction steps increased the yield only slightly. This phenomenon showed that, although acetone is the most common solvent used for the extraction of lichen metabolites [5,10,14,15,16,17,18,19,20], this extractant did not provide exhaustive extraction of UA from *C. uncialis*.

### 2.3. Polynomial Regression Modes Development

The four independent variables, including the camphor:thymol (X_1_) molar ratio, the NADES:sample/liquid:solid (X_2_) ratio, the temperature of extraction (X_3_), and the time of extraction (X_4_), were tested for the optimization of UA extraction from the lichen thallus (Table 2). However, as the temperature of extraction (X_3_) did not show a significant impact on the UA yield, this factor was removed from the model. The camphor:thymol ratio and, partially, the time (*p* = 0.051) exerted a significant linear and quadratic effect (Table 3). The first factor (X_1_) had a negative impact (−0.92), while X_4_ exerted a positive linear effect on the UA extraction yield. Surprisingly, not only the temperature of extraction (X_3_), but also the liquid:solid ratio (X_2_) did not exhibit a significant linear effect on the response. In turn, the X_2_X_4_ interaction effect was significant and X_1_X_2_ was close to significance (*p* = 0.059).

The constructed polynomial model was highly significant and exhibited good fitting statistics with a determination coefficient *R*^2^ over 0.71. Additionally, the predicted *R*^2^ was close to the adjusted *R*^2^ (difference not exceeding 0.15), which indicated that the model was adequate for prediction. In turn, a ratio of signal-to-noise above 4 (Adeq Precision) indicated that the developed model had an adequate signal level and high ability to predict properties within the whole designed space [43].

In conclusion, it should be highlighted that the NADESs used extracted UA very efficiently already in a single extraction process. A similar or even higher efficiency of extraction from various plant compounds using different NADESs compared to classically used extractants has already been repeatedly reported in other studies [32,33,34,36,40], and this experiment once again confirmed the effectiveness of NADESs as extractants.

### 2.4. Effect of Extraction Variables on the Usnic Acid Yield

Figure 4 presents the effect of the extraction parameters on the UA extraction yield. It was shown that the most favorable camphor:thymol ratio was 0.3 (Figure 4a,b), and the extraction yield decreased with the increasing camphor content in the mixture (Figure 4d). The positive effect of an increasing thymol content in the NADES has been demonstrated previously for the extraction of alkaloids from *Chelidonium majus* [34]. This effect of the high thymol content on the solubility and extraction efficiency of UA can be hypothesized to be related to the possibility of hydrogen bond formation between these molecules.

Our unpublished studies suggest that mixtures of UA and thymol melt at a much lower temperature than UA alone; however, the exact explanation of this phenomenon requires further research. In addition, it was shown that, with the decreasing camphor:thymol ratio, the obtained NADES had lower viscosity from 18.0 to 16.6 mPa/s for 1.5 and 0.3 molar rates, respectively, which is known to have a positive effect on the kinetics of the extraction process. The positive effect of high values of the liquid:solid ratio (Figure 4a) and time (Figure 4b) on extraction efficiency is consistent with Fick’s laws [44]. Figure 4c displays the interaction between the extraction time and the liquid:solid ratio. It was confirmed that, at low ratio values, time had no significant effect on UA extraction efficiency, while the extraction efficiency at high liquid:solid ratios increased significantly with the increasing extraction time. Also, these observations seem to be in agreement with Fick’s laws, according to which a larger amount of the extracted substance induces higher partial pressure and accelerates diffusion. The distance between the solute and the solvent and the contact area between the solute and the solvent have the same effect on the diffusion rate [44]. In this approach, the equilibrium concentration of UA in the extracted material and the extractant is established relatively quickly at low values of the liquid:solid ratio, and further extension of the extraction time has no significant effect on the extraction efficiency. When the values of the liquid:solid ratio are high, the effect of the volume of the extraction mixture (distance) must be compensated by increasing the extraction time, while a larger amount of the extractant increases the concentration gradient, which has a positive effect on the extraction process.

### 2.5. Optimization of the Extraction Process and Model Validation

The optimal set of factors that exerted a significant impact on the response were determined by maximization of the extraction yield. The results of the numerical optimization are presented in Table 4. Based on the 10 proposed solutions, it was found that a low value of the camphor:thymol ratio (average 0.32), a value of the liquid:solid ratio of 58.7, and 27.4 min of the extraction time ensured maximization of the extraction efficiency close to 91% of the total amount of UA present in the material.

Considering the numerical optimization, the following parameters: camphor:thymol ratio—0.3, liquid:solid ratio—60, and time—30 min were used in the experiment for confirmation of the model (Table 5). The experimental results and the predicted value corresponded well, with the mean experimental value falling within the confidence interval with deviations between 11.93 and 14.96.

## 3. Materials and Methods

### 3.1. Chemicals

β-Alanine, sucrose, L-proline, L-(+)-tartaric acid, citric acid, D-sorbitol, choline chloride, myristic acid, levulinic acid, lauric acid, DL-menthol (≥95%), (±)-camphor (≥95.5%), (−)-borneol (97%), lactic acid (90%), glycerol anhydrous, methanol (HPLC), acetonitrile (HPLC), UA (98%) were purchased from Sigma-Aldrich (St. Louis, MO, USA). Betaine was provided by Sigma Aldrich (Oakville, ON, Canada); glucose, maltose, and D-fructose were purchased from POCH S.A. (Gliwice, Poland); urea from Chempur (Piekary Śląskie, Poland); and propanediol from EcoSpa (Józefosław, Poland). Water was deionized and purified using ULTRAPURE Milipore Direct-Q^®^ 3UV–R (Merck, Darmstadt, Germany).

### 3.2. Lichen Material

Wild-growing *C. uncialis* lichen thalli were collected from Sobibór Forest, Poland (N 51°27′34.20″; E 23°36′32.90″). The samples were identified by Prof. Hanna Wójcik and deposited at the Department of Analytical Chemistry, Medical University of Lublin, Lublin, Poland (voucher specimen no. C.u.1/2022). The air-dried lichen was stored in a dry and dark place until use in the experiments. Then, it was mortar-ground to a homogeneous powder and sieved (1.6 mm sieve).

### 3.3. Preparation of NADESs

#### 3.3.1. Lipophilic NADESs

All lipophilic NADESs were obtained by heating the mixed components in a water bath without additional mixing. Initially, all NADESs were heated for 30 min at 50 °C. This made it possible to obtain liquids in thymol:camphor, thymol:lactic acid, thymol:lactic acid:choline chloride systems and in menthol: lactic, lauric, and myristic acid systems. The mixtures of borneol:thymol, menthol:thymol, and menthol:camphor were further heated for 30 min at 60 °C. To obtain a homogenous liquid of borneol:thymol, additional heating for 30 min at 70 °C was applied.

#### 3.3.2. Hydrophilic NADESs

The hydrophilic NADESs were dissolved in an excessive amount of water in a previously weighed round-bottomed flask. Then, the samples were evaporated under reduced pressure (10 mBa) in a rotary evaporator with a water bath (50 °C). Further, the flasks were placed in a desiccator for five days until constant weight was achieved. The mass difference between samples before and after water addition was considered as water bound in the NADES structure. The samples that crystallized during evaporation, drying in the desiccator, or storage at ambient temperature did not qualify for further experiments.

### 3.4. Determination of the Solubility of Usnic Acid in NADESs

An excess amount of UA was added to 1 mL of samples (lipophilic and hydrophilic NADESs) in 2 mL Eppendorf tubes. The tubes were transferred to an ultrasonic bath. Ultrasound at a frequency of 35 kHz (Sonorex RK 512 H, Bandelin, Berlin, Germany) was applied at ambient temperature in three cycles (each of 15 min).

The samples were centrifuged (15 min; 20,879 rcf, room temperature) after the ultrasonic bath. Afterward, the samples were allowed to settle for two weeks at 20 °C to attain equilibrium.

After that, the saturated UA solutions of NADESs were diluted in water (hydrophilic) or methanol (lipophilic solvents) in adequate ratios, depending on the UA content in the sample (1:1, 1:2, or 1:10). Subsequently, the samples were analyzed by HPLC. The concentrations of UA in the samples were transformed to the solubility value. The solubility of UA in one-component solvents was assessed similarly.

### 3.5. HPLC-PDA Analysis of UA

An EliteLaChrom chromatograph with a PDA detector and EZChrom Elite software (Merck, Darmstadt, Germany) was used for chromatographic analysis. The samples were analyzed on a C18 reversed-phase core–shell column (Kinetex, Phenomenex, Aschaffenburg, Germany) (10 cm × 4.6 mm i.d., 2.6 μm particle size) using a mixture of acetonitrile and water (60:40 *v*/*v*) with 0.025% of trifluoroacetic acid as the mobile phase at a temperature of 20 °C and an eluent flow rate of 1.2 mL·min^−1^. The sample injection volume was 10 μL. The UA retention time was determined as t_R_ = 8.42 min. The concentration of UA in the samples was determined from the area under the curve at 281 nm with a calibration curve obtained with the methanolic solution of UA (100–1000 μg/mL, y = 1E + 10x − 22,7314; *R*^2^ = 0.9998). All analyses were repeated three times.

### 3.6. Determination of the Total UA Content in Cladonia Uncialis Thallus

*Cladonia uncialis* thallus, pulverized and accurately weighed to 0.1000 g, was extracted ten times with acetone (10 × 1.5 mL) using an ultrasonic bath at 30 °C (10 × 15 min). After each extraction step, the sample was centrifuged, and the supernatant was poured into 2 mL volumetric flasks. The flasks were topped up with acetone and mixed thoroughly. The extracts obtained were analyzed by HPLC according to the methodology described in Section 3.5. The sum of the obtained results was taken as 100% of the content of UA in the thallus of *C. uncialis*.

### 3.7. Design of Experiment

The design of experiments methodology was used to investigate the influence of four independent numerical factors: camphor:thymol (X_1_) molar ratio, NADES:sample/liquid:solid ratio (X_2_), temperature of extraction (X_3_), and time of extraction (X_4_) on the yield of UA extraction from *Cladonia uncialis*. A Box–Behnken design was used for the generation of the design of the experiment (Table 2). Based on the obtained results, a polynomial model was developed and verified by checking the significance of the model using ANOVA and the normal distribution of residuals using the Shapiro–Wilk test (*p* < 0.05). Additionally, the adequacy of the models was checked by calculation of the fit statistics, including *R*^2^, adjusted *R*^2^, predicted *R*^2^, and adequate precision. Based on the developed model, numerical optimization was performed. Finally, the model was confirmed using the experimental (*n* = 3) and predicted values. All statistical calculations were performed using Statistica ver. 13.3.03 (Tibco Software Inc., Palo Alto, CA, USA) and Design Expert ver. 13 (Stat-Ease Inc., Minneapolis, MN, USA).

### 3.8. Preparation of Extracts

In order to perform the design of experiments, three different ratios (0.3, 0.875, 1.5) of the camphor:thymol mixture (Table 2) were prepared in the conditions described in Section 2.3. Air-dried and ground *Cladonia uncialis* was weighed directly into the Eppendorf tubes at three different weight levels (approximately 120 mg; 24.9 mg, and 41.25 mg). Then, 1.5 mL of various mixtures of camphor and thymol were added to the tubes. The samples were extracted in an ultrasonic bath for the assigned time and at a controlled temperature. The detailed conditions of all the conducted experiments are presented in Table 2. After the extraction procedures, the samples were centrifuged, and the supernatants of each extract were quantitatively transferred to vials and diluted in water (hydrophilic) or methanol (lipophilic NADESs) in a 1:1 ratio. Then, samples were analyzed by HPLC under the conditions described in Section 2.5.

### 3.9. Viscosity Measurement

Viscometric measurements of the camphor:thymol mixtures (in molar ratios 1.5, 0.875, 0.3) in a temperature range of 22–40 °C were performed using an IKA ROTAVISC lo-vi viscometer with an ELVAS-SP spindle, a small sample adaptor, and a temperature sensor (IKA^®^, Warsaw, Poland).

## 4. Conclusions

This study presents the results of UA solubility evaluation in 10 single-component solvents and 16 lipophilic and 51 hydrophilic NADESs, demonstrating the greatest potential of lipophilic NADESs as solvents for this compound. The solubility of UA is much higher in the mixtures of lipophilic NADESs than in single-component solvents and hydrophilic NADESs. Based on the use of the mixtures of thymol and camphor to optimize the UA extraction process, a polynomial model was constructed, which is highly significant and exhibits good fitting statistics with a determination coefficient *R*^2^ over 0.71. The model is suitable for prediction, has an adequate signal level, and has good ability to predict properties across the design space. Considering the numerical optimization, the following parameters: camphor:thymol ratio—0.3, liquid:solid ratio—60, and time—30 min were used in the experiment to verify the model. The experimental results and the predicted value corresponded well, with the mean experimental value falling within the confidence interval with deviations between 11.93 and 14.96. The model allowed optimization of the extraction procedure, whereby a single extraction yielded approximately 91% of the total UA content, while only 78.4% of UA were isolated in the single acetone extraction. The mechanistic explanation of this phenomenon requires further research, but it can be assumed that thymol and UA form a eutectic solvent.

## Figures and Tables

**Figure 1 molecules-28-05321-f001:**
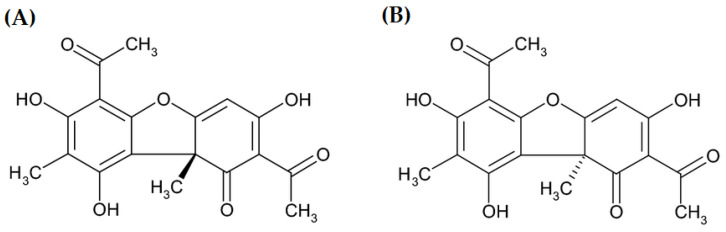
Two enantiomers of usnic acid: (**A**) (+)- usnic acid and (**B**) (−)-usnic acid.

**Figure 2 molecules-28-05321-f002:**
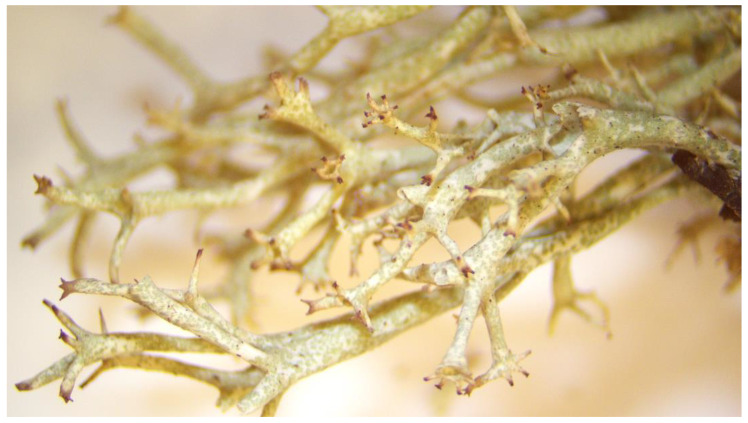
Microscopic image of the thallus of *Cladonia uncialis* (L.) Weber ex F.H. Wigg.

**Figure 3 molecules-28-05321-f003:**
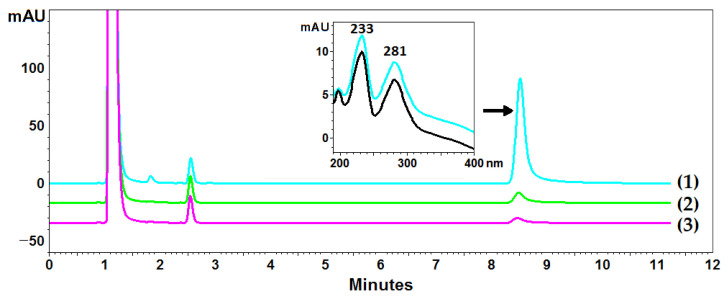
Example chromatograms of *Cladonia uncialis* extracts and the DAD spectrum of usnic acid. Lines (1), (2), and (3) correspond to chromatograms from subsequent extraction steps. The black line represents the spectrum of the UA standard, and the blue line represents the UA spectrum recorded in the extract.

**Figure 4 molecules-28-05321-f004:**
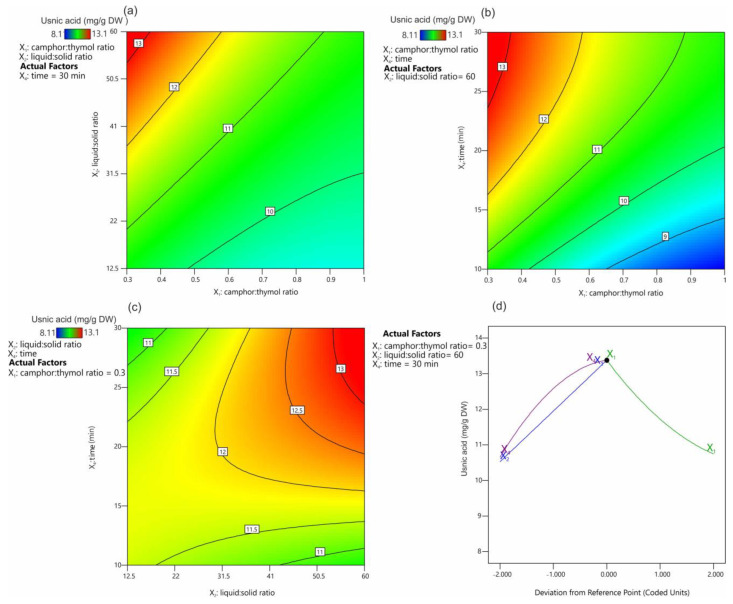
Contour plots (**a**–**c**) and trace (Piepel) plots (**d**) for the effect of extraction factors on the usnic acid yield.

**Table 1 molecules-28-05321-t001:** Usnic acid solubility in selected solvents at 20 °C.

One-Component Solvents		Solubility (mg/100 mL)
Ethyl acetate		656.72
Acetone		636.70
Dimethyl sulfoxide		345.52
Levulinic acid		55.50
1,3-Propanediol		41.74
Ethanol		32.76
Methanol		13.35
Lactic acid		5.07
Glycerol		1.38
Water		0.02
Lipophilic NADESs	Molar ratio	Solubility (mg/100 mL)
Thymol based		
Thymol:borneol	2:1	601.43
Thymol:borneol	3:1	1022.49
Thymol:lactic acid:water	1:2.79:1.55	401.43
Thymol:lactic acid:water	1:1.44:0.8	587.00
Thymol:lactic acid:water	1:2.7:1.5	319.10
Thymol:menthol	1:1	300.28
Thymol:menthol	2:3	498.85
Thymol:lactic acid:water:choline chloride	1:3.6:2:1	8.51
Camphor based		
Camphor:thymol	1:1	396.97
Camphor:thymol	1:4	904.50
Camphor:thymol	3:2	805.83
Camphor:thymol	3:7	841.30
Camphor:menthol	2:3	369.79
Menthol based		
Menthol:lactic acid:water	1:1.8:1	162.32
Menthol:lauric acid	2:1	113.98
Menthol:myristic acid	8:1	108.22
Mean value		465.125 (*p* < 0.0000) *

* Significantly different according to Mann–Whitney U test from the mean value (2.007) for hydrophilic NADESs (cf. Appendix A).

**Table 2 molecules-28-05321-t002:** Experimental factors and measured values of responses. The response was measured in triplicate ± SD.

Independent Variables				Responses
Molar Ratio Camphor:Thymo (X_1_)	Liquid:Solid Ratio (Volume/Mass) (X_2_)	Temp. (°C) (X_3_)	Time (min) (X_4_)	Usnic Acid (mg/g of Dry Lichen Thallus)
0.3	12.5	27.5	20	11.30 ± 0.64
1.5	12.5	27.5	20	11.42 ± 0.03
0.3	60	27.5	20	13.09 ± 0.27
1.5	60	27.5	20	10.41 ± 0.11
0.875	36.25	20	10	9.25 ± 0.2
0.875	36.25	35	10	10.35 ± 0.13
0.875	36.25	20	30	9.12 ± 0.27
0.875	36.25	35	30	9.82 ± 0.14
0.3	36.25	27.5	10	10.73 ± 0.18
1.5	36.25	27.5	10	9.76 ± 0.28
0.3	36.25	27.5	30	13.10 ± 0.19
1.5	36.25	27.5	30	10.46 ± 1.11
0.875	12.5	20	20	11.39 ± 0.24
0.875	60	20	20	9.45 ± 0.16
0.875	12.5	35	20	9.86 ± 0.29
0.875	60	35	20	10.15 ± 0.25
0.3	36.25	20	20	11.21 ± 0.35
1.5	36.25	20	20	10.43 ± 0.31
0.3	36.25	35	20	12.52 ± 0.22
1.5	36.25	35	20	10.62 ± 0.34
0.875	12.5	27.5	10	10.73 ± 0.36
0.875	60	27.5	10	8.11 ± 0.16
0.875	12.5	27.5	30	10.07 ± 0.53
0.875	60	27.5	30	11.26 ± 0.02
0.875	36.25	27.5	20	9.91 ± 0.17
0.875	36.25	27.5	20	10.58 ± 0.29
0.875	36.25	27.5	20	11.31 ± 0.49
0.875	36.25	27.5	20	11.05 ± 0.44
0.875	36.25	27.5	20	11.14 ± 0.28

**Table 3 molecules-28-05321-t003:** Results of fit statistics, analysis of variance, and estimated coefficients for usnic acid.

Fit Statistics				
*R* ^2^	0.7127			
Adjusted *R*^2^	0.6170			
Predicted *R*^2^	0.4692			
Adeq Precision	11.936			
Term ^1^	Coefficient estimate	Sum of squares	df	*p*-value
Model		24.39	7	0.0002
Intercept	10.92			
X_1_	−0.9162	12.97	1	<0.0001
X_2_	0.0740	0.0452	1	0.7591
X_4_	0.4083	2.00	1	0.0512
X_1_X_2_	−0.3985	1.87	1	0.0588
X_2_X_4_	0.9525	3.63	1	0.0111
X_1_^2^	0.3402	6.87	1	0.0010
X_4_^2^	−0.5690	2.23	1	0.0404
Residual		9.32	21	
Lack of Fit		8.55	17	0.3559
Pure Error		1.28	4	
Cor Total		34.22	28	

^1^ Variable code: X_1_—camphor:thymol ratio; X_2_—liquid:solid ratio; X_4_—time.

**Table 4 molecules-28-05321-t004:** Optimal response values computed by numerical optimization of estimated factors.

No.	Camphor:Thymol Ratio (X_1_)	Liquid:Solid Ratio (X_2_)	Temperature (X_3_)	Time (X_4_)	Response Usnic Acid Extraction Yield (mg/g DW)
1	0.304	57.548	27.5	28.434	13.241
2	0.314	58.780	27.5	25.580	13.108
3	0.300	56.705	27.5	26.134	13.127
4	0.320	59.081	27.5	28.906	13.252
5	0.327	58.932	27.5	27.046	13.130
6	0.305	59.460	27.5	24.717	13.120
7	0.342	59.728	27.5	28.923	13.168
8	0.309	58.512	27.5	25.271	13.100
9	0.362	59.986	27.5	29.992	13.104
10	0.308	58.228	27.5	29.341	13.282

**Table 5 molecules-28-05321-t005:** Predicted and experimental values (*n* = 4) at the set of optimal extraction factors: camphor:thymol ratio (X_1_)—0.3, liquid:solid ratio (X_2_)—60, time (X_4_)—30 min.

Response Variables	Predicted Mean Value	Experimental Mean Values	95% PI Low	95% PI High	RD (%)
Usnic acid (mg/g DW)	13.44	13.08	11.93	14.96	−0.36

RD—response deviation.

## Data Availability

All data are contained within this article. Raw datapoints from this study are available on request from the corresponding author.

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
