# Peer review of "Methodological Aspects of Green Extraction of Usnic Acid Using Natural Deep Eutectic Solvents"

_molecules, 2023, doi:10.3390/molecules28145321_

Round 1

Reviewer 1 Report

Kulinowska et al reports about green extraction of usnic acid using  natural deep eutectic solvents.. After close evaluation of the manuscript I suggest revision according to next comments:

1. The Introduction  does not provide state of the art in NADES extraction. NADES could be tuned for the extraction of  polar and non-polar compounds such as  alkaloids (https://doi.org/10.1016/j.chroma.2018.07.009), anthocyanins [(https://doi.org/10.1016/j.lwt.2021.111220), phenyletanes and phenylpropanoids (https://doi.org/10.3390/molecules25081826), steroidal saponins (https://doi.org/10.3390/molecules26072079 ), phlorotannins (https://doi.org/10.1007/s11094-019-01987-0), polysaccharides (https://doi.org/10.3390/proceedings2019029096), caratenoids and hydrophylic vitamins (https://doi.org/10.3390/molecules26144198).

2. In Table 1 authors have provided the data about solubility of US without statistics.The discussion about solubility should be based on statistical significance of differences in solubility.

3. Table 3: R^2 is quite low. whether the authors have tried a different mathematical model?

4. Sect 3.1: please provide purity of Lactic acid. How water content in lactic acid was counted for preparation of NADES?

5. Sect. 3.2: please provide information who have identified C. uncialis. Please provide number of voucher of specimens.

6. Figure 4: the legends inside of figure are too small. Please include all details in the Legend to Fig.5

7. In Conclusion: Does the solubility of UA in lipophilic NADES differ statistically significantly from the solubility of UA in hydrophilic NADES??

English language is fine, minor changes could be useful.

Author Response

We would like to thank the Reviewer for their comments, which we feel gave us the opportunity to strengthen the message of the manuscript. Please find a point-by-point response below.

Reviewer comment: The Introduction  does not provide state of the art in NADES extraction. NADES could be tuned for the extraction of  polar and non-polar compounds such as  alkaloids (https://doi.org/10.1016/j.chroma.2018.07.009), anthocyanins [(https://doi.org/10.1016/j.lwt.2021.111220), phenyletanes and phenylpropanoids (https://doi.org/10.3390/molecules25081826), steroidal saponins (https://doi.org/10.3390/molecules26072079 ), phlorotannins (https://doi.org/10.1007/s11094-019-01987-0), polysaccharides (https://doi.org/10.3390/proceedings2019029096), caratenoids and hydrophylic vitamins (https://doi.org/10.3390/molecules26144198).

Answer: As recommended by the Reviewer, we have mentioned the possibility of extracting both polar and non-polar compounds with NADES. We have also cited some of the publications suggested by the Reviewer . We apologize that we did not cite all the suggested sources, but this was due to the comments of the second reviewer, who suggested balancing the number of citations in the introduction and discussion.

Reviewer comment: In Table 1 authors have provided the data about solubility of US without statistics.The discussion about solubility should be based on statistical significance of differences in solubility.

Answer: The solubility values are physical properties that are specific to each system. These values do not show any deviations; therefore, the necessary conditions for conducting appropriate statistical tests are not met in this case. One of the ways to evaluate this type of data is through descriptive analysis (which has been performed).

Reviewer comment: Table 3: R^2 is quite low. whether the authors have tried a different mathematical model?

Answer: Several types of models and their components were tested. The presented model exhibits the highest possible level of fit and the highest possible prediction indicators.

Reviewer comment: Sect 3.1: please provide purity of Lactic acid. How water content in lactic acid was counted for preparation of NADES?

Answer: Dear Reviewer, thank you for this very important comment. The lactic acid used in this study had a concentration of 90%. We have completed this information in section 3.1. Taking this into account, we have corrected the information about the composition of the mixtures in Table 1. We have calculated the lactic acid content by multiplying the original value by 0.9. Thus, the molar ratio of menthol to lactic acid, which we originally stated as 1:2, is 1:1.8 after the correction. In addition, we have also given the water content, calculated as follows: original number of moles of lactic acid x 90.08 x 0.1 / 18.

Reviewer comment: Sect. 3.2: please provide information who have identified C. uncialis. Please provide number of voucher of specimens.

Answer: The relevant information has been added to Section 3.2.

Reviewer comment: Figure 4: the legends inside of figure are too small. Please include all details in the Legend to Fig.5

Answer: The legend of Figure 4 has been improved.

Reviewer comment: In Conclusion: Does the solubility of UA in lipophilic NADES differ statistically significantly from the solubility of UA in hydrophilic NADES??

Answer: Both mean values have been compared using the Mann-Whitney U test, and the results have been added to Table 1.

Reviewer 2 Report

The manuscript ID: molecules-2462506 entitled “Methodological aspects of green extraction of usnic acid using natural deep eutectic solvents” shows that the authors are maximizing the potential of NADES in the future utilization of usnic acid which is a compound with evidence-based multiple biological activities. The quality of the scientific English language is outstanding; the text is logically organized.

I have the following specific comments:

The manuscript cites 40 references, which are in line with the theoretical basis of the experiment, but most of them (as many as four-fifths) are mentioned in the Introduction part; it would be good to have some balance throughout the whole text, especially in the Discussion.

Table 1. is so long that it could be better placed in the supplementary material.

Regarding M&M, the extracts were prepared by centrifugation, and the authors reported the conditions in rpm without centrifuge radius, or it would be better to report rcf (in which case the study becomes more reproducible). In addition, one of the four factors in the development of the polynomial regression models is the molar ratio camphor: thymol; this can be somewhat confusing, since initially thymol: camphor was evaluated (Table 1).

Author Response

We would like to thank the Reviewer for their comments, which we feel gave us the opportunity to strengthen the message of the manuscript. Please find a point-by-point response below.

Reviewer comment: The manuscript cites 40 references, which are in line with the theoretical basis of the experiment, but most of them (as many as four-fifths) are mentioned in the Introduction part; it would be good to have some balance throughout the whole text, especially in the Discussion.

Answer: Thank you for this comment. We have tried to add some references in the discussion of the results (section 2.3). We hope that we have satisfied the Reviewer at least to some extent. This discussion is difficult, especially since no one has tested NADES as UA solvents to date.

Reviewer comment: Table 1. is so long that it could be better placed in the supplementary material.

Answer: We have moved part of Table 1 to supplementary materials. Only the solubility in monosolvents and lipophilic NADES has been left in the main body.

Reviewer comment: Regarding M&M, the extracts were prepared by centrifugation, and the authors reported the conditions in rpm without centrifuge radius, or it would be better to report rcf (in which case the study becomes more reproducible). In addition, one of the four factors in the development of the polynomial regression models is the molar ratio camphor: thymol; this can be somewhat confusing, since initially thymol: camphor was evaluated (Table 1).

Answer: We agree with the Reviewer that providing crf values is a correct and complete way to describe the methodology used. Therefore, we have removed the rpm value and provided crf. Thank you very much for this suggestion.

The notation throughout the manuscript has been standardized as camphor:thymol (paragraph 2.1.2 and Table 1).

Reviewer 3 Report

According to reviewed manuscript I suggest following revision:

Figure 1: Methodology in not needed here. In material and methods you already gave information about the column and method performing.

Introduction: You did not emphasized why UA to be used in industry, it must be isolated from lichen thallus?

Section 3.2. Size of herbal powder particles should be added.

Section 3.3 How was NADES prepared? By hating and mixing? You should extend this section with more details heating temperature, mixing, time.

Line 250-251 should be deleted.

Section 2.1 should be rewritten to be easier to follow and read. At least divide it to different paragraphs. Lot of information is given.

Line 113-114: However, the exhaustive extraction 113of UA with acetone was shown to be highly problematic and acetone did not seem to be 114thebest extractant of this compound (compared to the NADES used in this experiment). Why was acetone problematic, any special issue?

Table 5 looks bad. Delete n1-n4, write only mean vale and specific that 4 measurements were done.

Minor editing of English language required.

Author Response

We would like to thank the Reviewer for their comments, which we feel gave us the opportunity to strengthen the message of the manuscript. Please find a point-by-point response below.

Reviewer comment: Figure 1: Methodology in not needed here. In material and methods you already gave information about the column and method performing.

Answer: Thank you for this suggestion. We suppose it was about figure 3. As a rule, we provide methodological data under the chromatograms, but we have removed this information at the request of the Reviewer.

Reviewer comment: Introduction: You did not emphasized why UA to be used in industry, it must be isolated from lichen thallus?

Answer: We thank the reviewer for the apt suggestion. We have expanded this part of the introduction by adding relevant explanations. We hope that they will be satisfactory to the Reviewer.

Reviewer comment: Section 3.2. Size of herbal powder particles should be added.

Answer: The relevant information has been added.

Reviewer comment: Section 3.3 How was NADES prepared? By hating and mixing? You should extend this section with more details heating temperature, mixing, time.

Answer: The relevant information has been added.

Reviewer comment: Line 250-251 should be deleted.

Answer: This sentence has been removed as suggested by the Reviewer.

Reviewer comment: Section 2.1 should be rewritten to be easier to follow and read. At least divide it to different paragraphs. Lot of information is given.

Answer: We have divided this paragraph into three paragraphs. We hope this will be satisfactory to the Reviewer. At the same time, we thank you very much for this suggestion because the text has now become more readable.

Reviewer comment: Line 113-114: However, the exhaustive extraction of UA with acetone was shown to be highly problematic and acetone did not seem to be thebest extractant of this compound (compared to the NADES used in this experiment). Why was acetone problematic, any special issue?

Answer: This sentence has been modified by us. We admit that the term "problematic" was inappropriate here. Thank you for the right comment.

Reviewer comment: Table 5 looks bad. Delete n1-n4, write only mean vale and specific that 4 measurements were done.

Answer: The table has been modified in accordance with the Reviewer's recommendations.

Round 2

Reviewer 1 Report

he authors adequately responded to my comments and questions. The manuscript may be accepted in present form.